# Peer Presence Effect on Numerosity and Phonological Comparisons in 4th Graders: When Working with a SchoolMate Makes Children More Adult-like

**DOI:** 10.3390/biology10090902

**Published:** 2021-09-12

**Authors:** Leslie Tricoche, Elisabetta Monfardini, Amélie J. Reynaud, Justine Epinat-Duclos, Denis Pélisson, Jérôme Prado, Martine Meunier

**Affiliations:** 1IMPACT Team, Lyon Neuroscience Research Center, INSERM, U1028, CNRS, UMR5292, University Lyon, F-69000 Lyon, France; leslie.tricoche@etu.univ-lyon1.fr (L.T.); elisa.monfardini@inserm.fr (E.M.); amelie.reynaud@inserm.fr (A.J.R.); denis.pelisson@inserm.fr (D.P.); 2EDUWELL Team, Lyon Neuroscience Research Center, INSERM, U1028, CNRS, UMR5292, University Lyon, F-69000 Lyon, France; justine.epinat-duclos@inserm.fr

**Keywords:** social facilitation, social presence, peer presence, children, literacy, numeracy, reaction times distribution, ex-Gaussian model, diffusion model

## Abstract

**Simple Summary:**

The presence of others helps us when we are good or an expert at something and hinders us when we are bad or novice. Such social facilitation or inhibition is well-documented in adults, but much less in children despite the omnipresence of peers throughout education. To explore potential peer presence effects on children’s academic performance, fourth-graders performed basic numerical and language skills (typically mastered at their age) either alone or with a schoolmate. For comparison, the same was performed in adults. We found that a schoolmate’s presence enabled children to perform more like adults, with a better response strategy and faster and less variable response times than children tested alone. This provides research-based evidence supporting pedagogical methods promoting collective practice of individually acquired knowledge. Future studies pursuing this hitherto neglected developmental exploration of peer presence effects on academic achievements might have the potential to help educators tailor their pedagogical choices to maximize peer presence when beneficial and minimize it when harmful. The present study also paves the way towards a neuroimaging investigation of how peer presence changes the way the child brain processes cognitive tasks relevant to education.

**Abstract:**

Little is known about how peers’ mere presence may, in itself, affect academic learning and achievement. The present study addresses this issue by exploring whether and how the presence of a familiar peer affects performance in a task assessing basic numeracy and literacy skills: numerosity and phonological comparisons. We tested 99 fourth-graders either alone or with a classmate. Ninety-seven college-aged young adults were also tested on the same task, either alone or with a familiar peer. Peer presence yielded a reaction time (RT) speedup in children, and this social facilitation was at least as important as that seen in adults. RT distribution analyses indicated that the presence of a familiar peer promotes the emergence of adult-like features in children. This included shorter and less variable reaction times (confirmed by an ex-Gaussian analysis), increased use of an optimal response strategy, and, based on Ratcliff’s diffusion model, speeded up nondecision (memory and/or motor) processes. Peer presence thus allowed children to at least narrow (for demanding phonological comparisons), and at best, virtually fill in (for unchallenging numerosity comparisons) the developmental gap separating them from adult levels of performance. These findings confirm the influence of peer presence on skills relevant to education and lay the groundwork for exploring how the brain mechanisms mediating this fundamental social influence evolve during development.

## 1. Introduction

An unvarying feature of schools worldwide is that children are educated in the constant presence of peers, yet scientific research does not always take this social aspect of learning into account. The cognitive literature traditionally ignores it. The educational literature did explore classrooms’ peer compositions to determine the impact of peers’ attributes: same- or other-sex, same- or other-ethnicity, same or different cognitive and academic abilities, etc., on academic achievements [1,2,3,4,5]. Whether this knowledge can reliably be used to implement policies that improve academic outcomes remains, however, a matter of debate [6]. Very little is known, by contrast, about how peers’ mere presence, irrespective of their attributes, may, in itself affect academic learning and achievement. This is despite the fact that, contrary to children’s individual characteristics, peer presence can be relatively easily manipulated (minimized or maximized) in a variety of amenable ways (e.g., by adapting pedagogical methods, modulating visual privacy through classroom arrangements, or improving auditory privacy via inexpensive devices such as noise-cancelling headphones). 

A long history of social psychology studies showed that others’ presence changes adults’ performance, generally facilitating the expression of mastered responses while impairing the acquisition of novel ones [7]. This ubiquitous social influence, termed the social facilitation or inhibition phenomenon (SFI), occurs in humans and animals whenever others are executing the same task at the same time—coaction effect—but also when others are simply hovering nearby—mere presence or audience effect—[8,9]. SFI equally affects basic acts such as laughing or moving the eyes, physical skills such as running or dressing up, and cognitive functions such as memory or reasoning [9,10,11,12]. Strangers suffice to trigger SFI, but there is evidence that the effect increases with familiarity with the peer [13,14,15,16]. All the above findings hold, however, mostly for adults as only a small fraction of the extensive SFI literature concerns children. Children studies represent, for instance, 14/241 studies in Bond and Titus’ 1983 meta-analysis, and about 25/800 studies in Guerin’s 2010 book, i.e., 6% and 3%, respectively. Also, most of the available children SFI studies have focused on basic acts [17] and physical skills [18,19], thereby providing limited insights into the potential influence of the constant presence of peers on children’s academic achievements. Interest for a developmental approach of peer presence effect recently emerged, however, in the adolescent literature. Studies notably aimed to understand the negative influence of peers on adolescents’ reasoning [13,20] and decision-making [21,22,23,24]. Applying this approach to children could unveil both the positive and negative influences of peers’ mere presence on education. This knowledge might ultimately provide useful insights to educators about when to minimize, or on the contrary, maximize peers’ presence.

The present study’s first aim was to measure the extent to which peer presence might affect skills that are relevant to fundamental education in elementary school children. To address this question, we measured the change in performance on literacy and numeracy tasks produced by the presence of a coacting classmate in 8 to 10-year-old fourth-graders. We designed a task taping two skills, one (numerosity comparison) relevant to numeracy, and the other (phonological comparison) relevant to literacy. The reason for this choice was two-fold. Firstly, numerosity and phonological comparisons are simple skills typically mastered before 4th grade [25,26], and thus, should be facilitated by social presence. They should therefore provide insight into positive peer presence effects, which, unlike negative ones, remain poorly investigated in the developmental literature [27,28,29] despite their potential relevance to education. Secondly, numerosity and phonological comparisons possess distinct neural signatures [30,31]. The present study could thus lay the behavioral ground for a neuroimaging exploration of how peer presence can similarly facilitate two different cognitive processes controlled by distinct neural substrates. 

The present study’s second aim was to assess the development of the peer presence effect by comparing children to adults. To this aim, we tested college-aged young adults while they performed the same task, either alone or in the presence of a coacting familiar peer. We analyzed errors, reaction times (RTs) and effect sizes. RTs were also analyzed using (1) the ex-Gaussian model [32] to determine whether peer presence affected average performance, variability in performance, or extremes in performance, and (2) the diffusion model [33] to determine which, among the decision and nondecision (i.e., memory and motor) processes preceding a response, was affected by peer presence. 

## 2. Materials and Methods

### 2.1. Participants

We recruited 111 4th-graders (in 3 schools from the Lyon, France area), and 100 college-aged young adults (via social network advertising). Adults were compensated for their participation (10€). Eight children were excluded from the analyses because they suffered from reading disability, attention deficit, or anxiety disorder. Data were entirely missing from four children and three adults due to recording problems. These were excluded from the analyses. Data were partially missing (for one of the two trial types; see below) from two adults and one child. These data were retained (without imputation). This resulted in a final sample size of 196 cases, including 99 children (40 females, mean age 9.25 years, SD = 0.46, range: 8–10 years) and 97 adults (57 females, mean age 21.7 years, SD = 2.33, range: 18–32 years). Based on an a priori power analysis conducted through G*Power 3.1.9.7 (www.gpower.hhu.de accessed on 10 September 2021) with α = 0.05, an overall sample size of 82 cases was required to detect with adequate power (1-β = 0.80) the effect size of Wolf et al., (2015) Age × Condition × Difficulty interaction (η_p_^2^ = 0.09, d = 0.6), which revealed adolescents’ greater social inhibition of difficult relational reasoning relative to adults. The present sample of 196 subjects thus represents more than twice as many subjects as required to detect such an effect size and provides a power of 1−β = 0.99 to detect it.

### 2.2. Solitary Versus Social Testing 

Testing took place in a quiet room, either at school for children, or in the laboratory for adults. Half of the subjects underwent solitary testing (alone condition: children *n* = 48, 17 females; adults *n* = 48, 26 females), and the other half was tested in pairs of coactors (social condition: children *n* = 51, 23 females; adults *n* = 49, 31 females). For children, pairs were formed by their teacher according to known affinities among classmates. For adults, half of the recruited subjects were same-age (±2 years) pairs of friends, siblings, or significant others (data were pooled across the three types of partners, as preliminary analyses revealed no effect of this variable). In both conditions, the subject was facing the screen of a laptop computer with the two index fingers positioned over two keyboard response keys. When present, the familiar peer was seated next to the subject and performed the same task at the same time on a second computer. The experimenter always left the testing room after having instructed the subject(s) and started the appropriate computerized task. Each subject completed the experimental task plus a series of questionnaires as described in the next paragraphs. 

### 2.3. Task

To assess social facilitation, we probed two skills present before the age of 8 [25,26], i.e., nonsymbolic numerosity comparison and phonological comparison. Nonsymbolic numerosity comparison involves comparing quantities using approximate representations of numbers without relying on counting or numerical symbols [34]; it is a skill detectable as early as 6 months of age and it was argued to predict children’s later mathematics achievement [26,35]. Phonological comparison involves comparing the sound structure of words [25]; it is a skill practiced early on in preschool in France and it is predictive of children’s later ability to read [36]. Using Presentation^®^ (www.neurobs.com accessed on 10 September 2021), we programmed a task comprising 288 trials for adults. The task was downsized to 144 trials for children to minimize the duration of the interruption of their school day. For both adults and children, half of the trials involved numerosity comparison trials, the other half involved phonological comparison. Figure 1 illustrates the two trial types, and provides an overview of the measures and analyses included in the study.

In numerosity comparison trials, subjects were asked to decide which of two arrays of dots (presented one after the other) had the largest number of dots. Each dot array was presented for 800 ms, with a 200 ms white screen in between. The second dot array was then replaced by a red square for a duration varying randomly from 2800–3600 ms. Subjects were asked to respond as fast and as accurately as possible by pressing a keyboard key as soon as the second dot array appeared and before the red square turned off. One key was associated with “the first dot array has the largest number of dots” answer and another key with “the second dot array has the largest number of dots” answer (Figure 1A). In phonological comparison trials, subjects were asked to decide as fast and accurately as possible if two words presented one after the other rhymed or not. As in the numerosity comparison task, the two words were presented during 800 ms each and separated by a 200 ms white screen, and the subjects had to answer as soon as the second word appeared and before the red square turned off. One keyboard key was associated with “the two words rhyme” answer, whereas the other key was associated with “the two words do not rhyme” (Figure 1A). 

As in Prado et al. (2011, 2014) [30,31], each trial type comprised four levels of difficulty. For numerosity comparison trials, these levels corresponded to four different ratios of number of dots between the two arrays: 0.33, 0.50, 0.67, or 0.75 for difficulty Levels 1–4, respectively. The higher the ratio, the greater the difficulty. For phonological comparison trials, difficulty was defined by the congruence (Levels 1 and 2) or incongruence (Levels 3 and 4) of the words’ spelling and phonology. Easy Levels 1 and 2 respectively included pairs of words with identical orthography and phonology (o + p+, e.g., sac-lac [sak-lak]), or different orthography and phonology (o-p-, e.g., jeu-doux [ʒœ-du]). Difficult Levels 3 and 4 respectively included pairs of words with the same phonology but a different orthography (o-p +, e.g., dos-taux [do-to]), or the same orthography but a different phonology (o + p-, e.g., tapis-iris [tapi-iris]). 

To avoid carryover effects (changes in performance on the 2nd experimental condition due to the specifics of the 1st experimental condition), trial types (numerosity and phonological), and difficulty levels (1–4) were not presented successively. Rather, each block of eight trials comprised four trials of each type, one per difficulty level, appearing in pseudorandom order with no more than three consecutive trials of the same type. This design mixing numerosity and phonological comparisons entails switch costs (slower responses for switch trials than for nonswitch trials within blocks mixing the two), but these specific costs were found to be stable across age when general development-related slowing is taken into account [37]. They thus should not reduce the validity of the present developmental inferences. 

### 2.4. Stimuli

The dot arrays used for adults contained 12, 18, 24, or 36 dots and were created using the “multisensory condition” of Gebuis and Reynvoet’s generator, which controls for differences in cumulative surface areas and distribution of dot sizes to ensure that subjects’ response are based only on the number of dots [38]. The dot arrays used for children were simpler to obtain accuracy scores close to adult levels of performance. For approximately half of the children (*n* = 51), we used easier to discriminate arrays of 12, 18, 24, or 36 dots made with the less controlled “simple-sensory condition” of Dehaene et al., 2005 generator (www.unicog.org accessed on 10 September 2021). For the other children (*n* = 48), we used tightly controlled arrays generated with Gebuis and Reynvoet’s generator with half the number of dots used for adults, i.e., 6, 9, 12, or 18 dots. Children’s data were pooled across the two types of stimuli as preliminary analyses revealed no effect of this variable. Words contained 1 or 2 syllables and 3–8 letters, as in earlier studies [39,40]. Their frequency in French language according to New and Pallier’s dictionary [41] did not differ across the four levels of difficulty. Each word appeared only once during the task. 

### 2.5. Accuracy, Reaction Times, and Effect Sizes

As summarized in Figure 1B, using R (RStudio, v.1.0.136) or SYSTAT (v13), accuracy (i.e., the proportion of correct key presses relative to the total number of key presses) and manual reaction time (RT, i.e., the time separating the appearance of the second stimulus from the key press) were entered in two 2 × 2 × 2 × 4 ANOVAs with the between-subject factors Condition (Social, Alone) and Age (Children, Adults), as well as the within-subject factors Trial type (Numerosity comparison, Phonological comparison) and Difficulty (Level 1, Level 2, Level 3, Level 4). Posthoc comparisons appropriate to factors that do not interact [42] were conducted through two-sample Student’s *t* tests with the Bonferroni adjustment for multiple comparisons. For RTs, a supplementary 2 × 2 × 2 ANOVA with the within-subject factor Switch (Yes, No), and the between-subject factors Age and Condition was performed to determine whether switch costs (the switch-minus-nonswitch trials difference in RT) were affected by age or peer presence. Effect sizes were reported as partial eta squared values (η_p_^2^) for each ANOVA. In addition, peer presence effect size in children and adults was compared using common language effect size (CL) and Cohen’s d_s_ [43]. CL was calculated by dividing the difference between the means for the Alone and Social conditions by the square root of the sum of their variances, and then determining the probability associated with the resulting z score. It gives the probability that a score selected randomly from one condition will be greater than a score selected randomly from the other condition. Cohen’s d_s_ was calculated by dividing the difference between the means for the Alone and Social conditions by the standard deviation pooled across the two conditions. It converts the estimated effect to a standardized estimate in SD units. A commonly used interpretation is to refer to effect sizes as small for d_s_ = 0.2, medium for d_s_ = 0.5, and large for d_s_ = 0.8 [43]. 

### 2.6. RT Distributions

As empirical RT distributions are usually not normally distributed but rather positively skewed, mean and variance in these cases do not fully describe the distribution [44,45]. RStudio was therefore used to compute group RT distributions (compiling correct trials across all subjects and difficulty levels) for each condition, age, and trial type. As summarized in Figure 1B, we used Kolmogorov–Smirnov (K-S) tests to assess the Condition effect on the RT distributions, followed by two complementary analyses.

Firstly, because Condition did change RT distributions, an ex-Gaussian decomposition of individual RTs was performed using the ‘retimes’ package (v.0.1–2, 2013) of R to determine whether peer presence affected the subjects’ average RT, variability in RT, or extremes in RT. The ex-Gaussian distribution results from the convolution of a Gaussian and exponential distribution, thereby generally providing excellent fits for skewed RT distributions [44,46]. It estimates three parameters, mu, the mean of the Gaussian component (µ models shorter/longer mean RT), sigma, the standard deviation of the Gaussian component (σ models the symmetrical variability around μ), and tau, the mean and standard deviation of the exponential decay (τ models the tail of extremely long RTs). Changes in μ result in a left- or rightward shift of the distribution, changes in σ result in a widening or narrowing of the overall distribution, and changes in τ result in stretching of the right tail of the distribution [44]. The three parameters, calculated for each subject and trial type, are provided in Appendix A. Individual plots of actual RT distribution superimposed with the model distribution of a sample of 1000 RT values iteratively drawn based on the model estimated parameters are provided in Appendix A. The three parameters were then analyzed using 2 (Condition) × 2 (Age) × 2 (Trial type) ANOVAs and Bonferroni adjusted Student’s *t* tests. 

Secondly, because the adults’ group RT distribution took the form of a bimodal distribution, indicative of two discrete response strategies—a faster one and a slower one [47,48]—we examined individual RT distributions to classify subjects as either fast or slow responders. Cochran–Mantel–Haenszel tests (CMH χ2, with the continuity correction) were conducted for each age and trial type to determine peer presence effect on the proportions of fast and slow responders. In addition, two 2 (Responder type: fast, slow) × 2 (Age) × 2 (Condition) ANOVAs (one per trial type) were performed on percent correct responses averaged over difficulty levels to determine whether fast and slow responders differed in accuracy.

### 2.7. Diffusion Modeling 

As summarized in Figure 1B, a diffusion model was then used to determine which decision process (among those leading to a response) was influenced by peer presence. Diffusion models were developed to explain simple, two-choice decision processes for which relatively rapid response decisions are required [33,49,50]. It assumes that information is accumulated via a noisy information accumulation process until a decision criterion is met, at which point a response is initiated. The diffusion model uses RT distributions of both correct and incorrect responses to estimate three parameters: (1) the drift rate (v; an index of how quickly and efficiently an individual can accumulate information to inform his/her response decision, which is theoretically linked to neural signal-to-noise ratio); (2) the boundary separation or threshold (a; how “certain” a person needs to be before committing to a response, or their speed-accuracy tradeoff setting), and (3) the nondecision time (t0; the time it takes to complete all other information processes, which, in our paradigm, mainly include the working memory process necessary to compare the two successively presented stimuli and the motor process necessary for the preparation of the response). 

We fit the model to each individual’s RT data, as in previous studies [32,51]. We suppressed fast guesses by removing the first centile of the group distribution of RTs and extreme outliers by removing RTs exceeding four standard deviations. Such suppressions concerned one or a few trials in more than 70% of subjects and represented 4.24% ± 0.01 to 10.88% ± 0.01 of the data collected per age, condition, and trial type. The software fast-dm was used to estimate v, a, and t0, as well as their intertrial variability, szr, sv, and st0, respectively. The starting point (zr) remained constant at 0.5 due to the absence of decision bias in our task (the two responses were equally probable). The model was fit to each individual data using Kolmogorov–Smirnov criterion, as Chi–Square criterion was not applicable, particularly for children, who performed less than 200 trials [51,52]. Fitting indexes and estimated parameters values are provided in Appendix A. They show that, overall, the data for each subject were well fitted. Individual distributions, superimposing actual data and model data, plotted using the ‘rtdists’ package (v.0.8–3, 2018) of R developed by Singmann and collaborators, are provided in Appendix A. For each task, 2 (Age) × 2 (Condition) × 2 (Responder type) ANOVAs were conducted on each of the 6 parameters estimated by the diffusion model (a, v, t0, szr, sv, st0). 

### 2.8. Questionnaires

As summarized in Figure 1B, subjects completed four questionnaires.

#### 2.8.1. Pairs’ Relationship Quality

The single-item, seven-point IOS (Inclusion of Other in the Self) scale was used to quantify the closeness of the relationship within each dyad of coactors [53]. The IOS scale presents seven pairs of circles (one labeled “self,” the second labeled “other”) whose overlap ranges from none to almost complete. The subjects selected the pair that best described their relationship with their coactor. Scores of four or more are considered as reflecting close relationships [54,55]. 

#### 2.8.2. Personality and Self-Efficacy

Earlier studies suggest that a positively oriented personality [56], or a high self-efficacy (i.e., a strong belief in one’s ability to perform a specific task, Sanna, 1992), may lead to greater sensitivity to peer presence. Therefore, we evaluated these two individual characteristics to control for their potential confounding effect on the differences observed with vs. without a peer. Self-efficacy (SE) was evaluated using the French version of the Skills Perceptions in Life Domains Scale [57], which evaluates adults’ SE in the leisure, interpersonal relations, education and general life contexts, or of its simplified version for children [58]. Personality was evaluated using the French versions of the Big Five Inventory for adults [59] and children [60]. These provide a self-assessment of the five basic personality dimensions of Extraversion/Energy, Agreeableness, Conscientiousness, Neuroticism/Emotional instability, and Openness/Intellect. 

#### 2.8.3. Resistance to Peer Influence

As susceptibility to peer presence might be related to self-reported resistance to peer influence (RPI), all present subjects completed the French version (courtesy of T. Paus) of Steinberg and Monahan’s 2007 4-point RPI scale whose higher scores indicate greater RPI [61]. Note that 9 subjects (8 adults, 1 child) were excluded from this analysis due to incorrect completion of the questionnaire (selection of two answers per item instead of only one). 

Questionnaire scores were compared using Age × Condition ANOVAs, Student’s *t*-tests or correlation tests as appropriate. The significance level was set at *p* < 0.05 for all analyses.

## 3. Results

### 3.1. Age, Trial Type, and Difficulty Effects 

Figure 2 illustrates accuracy and RTs as a function of Age, Condition, Trial type, and Difficulty. The 2 × 2 × 2 × 4 ANOVA on accuracy (Figure 2A) showed a main effect of Age (F(1,189) = 98.75, *p* < 0.001, η_p_^2^ = 0.34), indicating higher accuracy in adults than in children. There was also a main effect of Trial type (F(1,189) = 10.45, *p* = 0.001, η_p_^2^ = 0.05), reflecting higher accuracy for numerosity comparison than phonological comparison. As predicted, the main effect of Difficulty was significant as well (F(3,567) = 291.88, *p* < 0.001, η_p_^2^ = 0.61). It was, however, qualified by a three-way Age × Trial type × Difficulty interaction (F(3,567) = 22.40, *p* < 0.001, η_p_^2^ = 0.11), which indicated that only for phonological comparisons (where children were tested with the same stimuli as adults) was the accuracy decline accompanying increasing difficulty sharper in children than in adults. For numerosity comparisons (which used simpler stimuli in children than in adults), the accuracy declines accompanying increasing difficulty in children and adults were indistinguishable, suggesting that simplifying children’s stimuli successfully equated task difficulty across the two age groups. 

The same patterns were observed for RTs (Figure 2B); namely, the 2 × 2 × 2 × 4 ANOVA yielded a main effect of Age (with faster responses in adults than in children, F(1,189) = 52.34, *p* < 0.001, η_p_^2^ = 0.22), a main effect of Trial type (with faster responses for numerosity than phonological comparison, F(1,189) = 273.94, *p* < 0.001, η_p_^2^ = 0.59), and a main effect of Difficulty (F(3,567) = 187.86, *p* < 0.001, η_p_^2^ = 0.50) qualified by an Age × Task × Difficulty interaction, indicating that the RT slow-down accompanying increasing difficulty was greater in children than in adults for phonological, but not numerosity, comparisons (F(3,567) = 7.54, *p* < 0.001, η_p_^2^ = 0.04). 

### 3.2. Peer Presence Effect

The global 2 × 2 × 2 × 4 ANOVA on RTs yielded a main effect of Condition (F(1,189) = 9.00, *p* = 0.003, η_p_^2^ = 0.05) revealing a social facilitation, i.e., faster responses in the Social than in the Alone condition. There was no Age × Condition interaction (F(1,189) = 1.40, *p* = 0.24), indicating that the overall magnitude of this RT speedup did not significantly differ across ages. Bonferroni adjusted Student’s *t* tests unveiled, however, significant Alone-minus-Social differences only in children. These significant differences concerned difficulty levels 2 and 3 of numerosity comparisons and difficulty level 4 of phonological comparisons (mean and 95% CI: Num2 = 223 ms (86,360), Num3 = 257 ms (121,393), Pho4 = 292 ms (123,462); all *p* values < 0.05, see asterisks in Figure 2B). 

Bonferroni-adjusted Student’s *t* tests revealed, in addition, that, for phonological comparison, children’s RTs in peer presence, however improved, generally remained below adult levels of performance (Children-Social minus Adults-Alone difference means and CI: Pho1 = 251 ms (111,391), *p* = 0.01; Pho2 = 377 ms (230,523), *p* < 0.001; Pho3 = 240 ms (60,419), *p* = 0.12; Pho4 = 333 ms (162,503), *p* = 0.005). For numerosity comparison, by contrast, children’s improved RTs in peer presence fell within the range of adults’ RTs in the alone condition (Children-Social minus Adults-Alone difference: Num1 = 135 ms, Num2 = 77 ms, Num3 = 11 ms, Num4 = 116 ms, all *p* values > 0.67; see the “=” sign in Figure 2B). This suggested that peer presence enabled children to at least partially fill up their developmental lag relative to adults (in the case of phonological comparisons more demanding for children than for adults), and at best, fully compensate their developmental lag (in the case of numerosity comparisons whose difficulty was successfully equated across age groups).

Note that peer presence improved RTs at minimal cost in accuracy. Compared to that of the Alone Condition, percent correct responses in the Social condition over the two trial types and four difficulty levels dropped by no more than 0.1 to 5.4% in children (mean 2.8%) and 0.0 to 4.7% in adults (mean 1.2%). Accordingly, the global 2 × 2 × 2 × 4 ANOVA on accuracy yielded a marginal main effect of Condition (F(1,189) = 4.05, *p* = 0.046, η_p_^2^ = 0.02) with no interaction between Condition and Age or other factors.

Finally, the supplementary 2 (Switch) × 2 (Age) × 2 (Condition) ANOVA showed an also marginal main effect of Switch (F(1,192) = 4.01, *p* = 0.047, η_p_^2^ = 0.02) with no interaction with Age or Condition. Switch costs (i.e., the additional time needed to respond when trial type changed) amounted to 23 ms in both children and adults tested alone, and to 24 and 21 ms in children and adults (respectively) tested with a peer. This confirmed the stability of these specific costs over development [37]. It indicated in addition that peer presence did not hasten (or impede) the flexibility process specific to switch trials. 

### 3.3. Effect Sizes 

Figure 3 illustrates the size of peer presence effects on reaction times averaged across difficulty levels. Effect sizes were greater in children than in adults for both trial types. For numerosity comparison, the average response of subjects tested with a peer was 217 ms (17%) faster in children and 124 ms (13%) faster in adults than the average response of subjects tested alone. The CL effect size indicated that, for each randomly selected pair, the chance that a subjec*t* tested with a peer responded faster than a subjec*t* tested alone was 67% for children and 60% for adults. Cohen’s d_s_ estimated peer presence effect as a medium effect of 0.61SD in children and a small effect of 0.38SD in adults (Figure 3). For phonological comparison, the average response in peer presence was 199 ms (12%) faster in children, and 74 ms (7%) faster in adults. The chance that a subjec*t* tested with a peer responded faster that a subjec*t* tested alone was 64% for children and 56% for adults. Cohen’s d_s_ reached a medium size of 0.51 in children and a small size of 0.21 in adults. 

### 3.4. Group RT Distributions

Figure 4 illustrates group RT distributions compiling correct trials across all subjects and difficulty levels. In both the Alone and Social conditions, children’s distribution was unimodal, whereas adults’ distribution was bimodal. For numerosity comparison, all adults taken together showed a 1st peak around 465 ms and a 2nd peak around 1107 ms with a trough latency around 910 ms. For phonological comparison, they showed a 1st peak around 624 ms and a 2nd peak around 1119 ms with a trough latency around 934 ms. K-S tests indicated that peer presence significantly modified group RT distribution for both numerosity and phonological comparison and in both children and adults (Children: numerosity comparison, D = 0.2, phonological comparison, D = 0.16; Adults, D = 0.13, and D = 0.09, respectively; all *p* values < 0.001).

Figure 5 illustrates the ex-Gaussian parameters, µ (average RT), σ (variability in RT), and τ (extremes in RT) obtained for each trial type. The 2 (Age) × 2 (Trial type) × 2 (Condition) ANOVAs yielded main effects of Age and Trial type for all three parameters (all Fs(1,190) ≥ 7.91, all ps ≤ 0.005, η_p_^2^ ≥ 0.20) as µ, σ, and τ were longer in children than in adults, and longer for phonological than for numerosity comparisons. For µ, there was in addition a main effect of Condition (F(1,190) = 11.06, *p* = 0.001, η_p_^2^ = 0.06) without interaction with other factors. For σ, there was a main effect of Condition (F(1,190) = 4.19, *p* = 0.04, η_p_^2^ = 0.02) qualified by a Condition × Age interaction (F(1,190) = 4.10, *p* = 0.04, η_p_^2^ = 0.02). For τ, there was no effect of Condition, and no interaction. 

Together these findings indicated that, for both trial types, peer presence shortened adults’ average RT (i.e., produced a leftward shift of the distribution; see Figure 4) whereas, in children, it both shortened the average RT and reduced the variability in RT (i.e., produced a leftward shift plus a narrowing of the distribution; see Figure 4). At neither age did peer presence affect the right tail of the distribution; that is, peer presence did not change the frequency of the extremely slow RTs. 

Bonferroni-adjusted Student’s *t* tests indicated that peer presence effects on µ and σ reached statistical significance only in children, and only for numerosity comparison (Alone-minus-social difference for children: µ-Num, 216 ms (90,342), *p* = 0.003; σ-Num, 56 ms (19,92), *p* = 0.03 see asterisks in Figure 5). Bonferroni-adjusted Student’s *t* tests indicated, in addition, that, for numerosity comparison, children’s average RT, µ, once improved by peer presence, no longer differed from the average RT of adults tested Alone (Children-Social minus Adults-Alone: 9 ms (–135,118), *p* = 1.0). 

### 3.5. Individual RT Distributions 

As illustrated in Figure 6, adults’ individual RT distributions were unimodal indicating that the 1st and 2nd peaks of the adults’ bimodal group distribution corresponded to different subjects favoring distinct response strategies (a fast or a slow one). Specifically, within the adults tested alone, half of the subjects (51%) were fast responders (i.e., with a RT peak < to the population trough latency; blue distributions in Figure 6), while the other half (49%) were slow responders (i.e., with a RT peak > to the population trough latency; orange distributions in Figure 6). For comparison, we similarly distinguished children whose peak latency was < vs. > to the adult trough latency plus the mean difference between adult and children RTs (that is, 1166 ms for Numerosity comparison and 1365 ms for phonological comparison; see Figure 4). In the Alone condition, children resembled the adults with about a half of fast responders (44% for Numerosity and 47 % for phonological comparison). CMHχ^2^ tests revealed that peer presence effect on the proportions of fast/slow responders differed with age (Numerosity: CMHχ^2^(1) = 12.96, *p* < 0.001; phonological comparison: CMHχ^2^(1) = 5.19, *p* = 0.02). Children showed sharp increases of the proportion of fast responders, up to 82% for Numerosity comparison (χ^2^(1) = 13.81, *p* < 0.001), and 70% for phonological comparison (χ^2^(1) = 4.52, *p* = 0.03), while adults showed only slight increases of fast responders up to 65% for Numerosity comparison and 61% for phonological comparison (χ^2^(1) = 1.46, *p* = 0.23, and χ^2^(1) = 0.82, *p* = 0.36, respectively).

Responder type × Age × Condition ANOVAs run on the average percent correct responses for each trial type showed no main effect of Responder type (Numerosity comparison: F(1,186) = 1.62, *p* = 0.20; phonological comparison: F(1,186) = 0.33, *p* = 0.56) with no interaction with the other factors. This indicated that accuracy was comparable in fast responders (who possibly anticipated part of the comparison process before the onset of the second stimulus) and slow responders (who possibly waited until the second stimulus onset to initiate the comparison process). Together, the above findings suggest that there were two equally efficient response strategies to solve the task, a fast one and a slow one. The two strategies were equally present in children or adults when they were tested alone, but not when they were tested with a peer. There, subjects favoring the fast response strategy over the slow one became the majority, and this strategy optimization produced by peer presence was more marked in children than in adults. 

### 3.6. Diffusion Modeling 

The main results are illustrated in Figure 7. Analyses of the diffusion model parameters using 2 (Age) × 2 (Condition) × 2 (Responder type) ANOVAs revealed a main effect of Age on all three modeled parameters. As illustrated in Figure 7A, relative to children, adults expectedly showed better decision parameters, with a faster drift rate v and a lower threshold a, as well as a better nondecision parameter, i.e., a lower t0 (numerosity comparison: F(1,186) = [16.5;32.58], *p*’s < 0.001 for a, v and t0, η_p_^2^ = [0.08;0.15]; phonological comparison: F(1,188) = [121.62;163.78], *p*’s < 0.001 and η_p_^2^ = [0.39;0.47] for v and t0, *p* = 0.43 for a). Trial-by-trial variability was also generally smaller in adults (Numerosity comparison: F(1,186) = [3.76;18.52], *p* = 0.05 and *p* < 0.001 for sv and st0 respectively, η_p_^2^ = [0.02;0.09]; phonological comparison: F(1,188) = [4.33;42.47], *p* = 0.04 and *p* < 0.001 for szr and st0 respectively, η_p_^2^ = [0.02;0.18]). Figure 7B shows, in addition, that fast responders differed from slow responders by their shorter and less variable nondecision time t0 (for both trial types, Numerosity comparison: t0: F(1,186) = 444.72, *p* < 0.001, η_p_^2^ = 0.71; st0: F(1,186) = 38.37, *p* < 0.001, η_p_^2^ = 0.17; phonological comparison: t0: F(1,188) = 490.2, *p* < 0.001, η_p_^2^ = 0.72; st0: F(1,188) = 34.02, *p* < 0.001, η_p_^2^ = 0.15). Posthoc analyses revealed no other differences between fast and slow responders, whatever the age, trial type, or condition. Finally, Figure 7C,D show that nondecision time t0 was the only parameter affected by Condition (Numerosity comparison: F(1,186) = 40.4, *p* < 0.001; phonological comparison: F(1,188) = 24.59, *p* < 0.001). t0 was reduced in the Social relative to the Alone condition. There was no Condition × Age effect on any parameter, indicating that the peer presence effect on t0 was comparable in children and adults. This was confirmed by posthoc analyses of t0 in the Alone vs. Social conditions (*p*’s ≤ 0.005 for both children and adults in both tasks). In addition, a correlation test between t0 and the mean RT revealed a significant positive correlation for both children and adults across both trial types and both conditions (Pearson, t = [12.18:19.27], all *p*’s < 0.001 for all r’s > 0.86). 

The diffusion model analysis therefore suggests that peer presence did not affect the decision parameters v and a, respectively modeling how fast and confidently subjects make their decision. Peer presence selectively shortened the nondecision parameter t0, which models all other information processes including, in our paradigm, the memory process necessary to compare the two successively presented stimuli and the motor process necessary to prepare the response. This t0 shortening enabled subjects to adopt the faster of the two response strategies adapted to solve the present task.

### 3.7. Questionnaires

Both adult and children coactors reached IOS scores greater than the 4/7 score considered as reflecting close relationships (adults: 5.9 ± 0.1; children: 5.2 ± 0.3), thereby confirming that, as intended, the coactor was a familiar peer (rather than a stranger) for both age groups. IOS scores in the adult pairs of friends, siblings, or significant others did exceed, however, the IOS scores of the children pairs of classmates (t = 2.39, df = 70.68, *p* = 0.02). As familiarity was reported to exacerbate peer presence effects [13,14,62], this slightly greater closeness of the socially tested adults possibly maximized the social facilitation observed in adults. 

*T*-tests on scores in the personality and self-efficacy questionnaires revealed no difference between the subjects tested in the Alone and Social conditions except for a slightly but significantly greater Extraversion/Energy score of the adults in the Social condition compared to that of the adults in the Alone condition (27.14 ± 0.79 vs. 24.63 ± 0.78: t = −2.27, df = 92.98, *p* = 0.03). As extraverted individuals tend to show greater peer presence effects [56], this slight bias of the socially tested adults towards extraversion possibly also maximized the social facilitation observed in adults. 

Finally the Condition × Age ANOVA on RPI scores did confirm earlier reports [61] of a lower (self-reported) resistance to peer influence in children (2.77 ± 0.04) than in adults (3.02 ± 0.04, Age effect: F(1,181) = 18.34, *p* < 0.001), associated with a positive correlation between RPI scores and age (Pearson, t = 4.42, *p* < 0.001, r = 0.31). It ascertained, in addition, that there was no main effect of Condition on RPI scores (Alone vs. Social: (F(1,181) = 0.74, *p* = 0.39). 

## 4. Discussion

Numerosity and phonological comparisons, two cognitive skills mastered early during education, were performed faster by 8 to 10-year-old 4th graders in the presence of a classmate than alone. This social facilitation was at least as important as that seen in adults. Ex-Gaussian decomposition of RTs revealed no peer presence effect on the extremely slow RTs thought to reflect lapses of attention [63]. Rather, peer presence reduced children’s RTs average and variability. Longer and more variable RTs are two well-known markers of cognitive immaturity during development [33,64]. By reducing both RTs average and variability, peer presence enabled children to compensate for their developmental lag, virtually completely for unchallenging numerosity comparisons (which were equally difficult for children and adults), and partly for demanding phonological comparisons (which were more difficult for children than for adults). Peer presence seemed, in addition, to boost children’s capacity to develop a response strategy. RT distributions in adults were indicative of two distinct response strategies [47,48], equally successful in terms of accuracy, although one was about 600 ms faster than the other. Based on debriefing, fast responders possibly optimized time by forming a preliminary opinion about their future response right from the 1st item onset, whereas slow responders possibly formed an opinion only after the 2nd item onset. The diffusion model analysis indicated that strategies did not change the decision processes determining how fast and confidently a decision is made. Rather, fast responders gained time by anticipating nondecision processes such as the comparison of the two items in working memory and/or preparation of the motor response. Unlike adults’, children’s RTs were not orderly organized into two well-defined strategies. Peer presence nevertheless allowed children to surpass adults’ organization as 70–82% of socially tested children adopted the optimal fast response strategy compared to only 61–65% of the socially tested adults. The diffusion model analysis indicated that, there too, peer presence made children more closely resemble adults, this time, by speeding up their nondecision processes.

As evoked in the Introduction, the abilities to compare numerical quantities or language sounds are two early developing skills that are foundational to the growth of math and reading skills in children. Both are arguably mastered by 4th grade, so social psychology consensual rule that others’ presence helps us execute mastered tasks predicted that both should be socially facilitated. We could not be certain, however, that this would be the case here, as our paradigm mixed trials of unequal difficulty. Indeed, whether a behavior is socially facilitated or inhibited eventually depends on the overall level of difficulty of the task it is embedded into. Bond showed as early as 1982 that an observer’s presence impairs the learning of three simple items if they are mixed with 10 difficult ones, and does not impair the learning of three complex items if they are mixed with 10 easy ones [65]. Likewise, we found earlier that simple (pro-)saccades (to the target), which are socially facilitated when performed by themselves, are socially inhibited when mixed with difficult antisaccades (away from the target) [10], whereas we show here that difficult phonological comparisons are socially facilitated when mixed with easy numerosity comparisons. Together, the present and earlier [10] findings suggest that two (potentially related) factors might help predict the final, overall level of difficulty of mixed protocols: mixing costs and neural substrates. Mixing pro- and antisaccades indeed came at a higher cost than mixing numerosity and phonological comparisons. We observed an accuracy loss in mixed blocks (compared to separate blocks) of 14% for saccades [10], compared to only 3–6% for numerosity and phonological comparisons (Prado et al., 2014 2nd to 7th graders compared to the present 4th graders: 3–4% loss; Prado et al., 2011 adults compared to the present adults: 5–6% loss). Higher mixing costs might be linked to the fact that pro- and antisaccades compete for a unique neural resource (the brain eye fields [66]), whereas numerosity and phonological comparisons possess distinct neural substrates (the intraparietal sulcus and posterior superior parietal lobule for the former, and the inferior frontal and middle temporal gyri for the latter [30,31]). In any case, the present and earlier results converge to suggest that the benefits of peer presence could be exploited in the classroom by embedding a low-to-substantial proportion of demanding items within unchallenging ones. 

As also evoked earlier, the long history of social psychology SFI studies predominantly concerned adults. In children, available studies often highlighted the positive influences of peers on basic acts and physical activities. In adolescents, available studies rather emphasized the negative influences of peers on cognitive skills. The present findings provide evidence that peer presence effects extend to the cognitive domain not only in adolescents, but also in children. They also underscore that sensitivity to peer presence in the cognitive domain is not always a liability, and can, at times, be adaptive. To improve our understanding of peers influences on academic achievements, future studies will therefore need to encompass both their harmful and their beneficial consequences throughout the entire course of education, from childhood to early adulthood [67,68]. 

The developmental trajectory of SFI remains unknown, as few studies compared peer presence effects across different ages. One study compared completion of jigsaw puzzles in children aged 5 and 8 years, and early adolescents aged 11 years [69]. Only the oldest group showed a performance impairment in the presence of an unfamiliar peer. Another study compared nonverbal reasoning in 10-year-old children vs. 13-year-old early adolescents with behavioral difficulties [70]. Both groups were slower to complete the task in the presence of a classmate than when alone, but only the oldest group committed in addition more errors in peer presence. A more recent study compared relational reasoning in early adolescents aged 10–14, late adolescents aged 15–18, and young adults aged 22–35 [13]. Adolescents, but not adults, showed poorer performance in the presence of a friend than in the presence of the experimenter, and this impairment was the most consistent across task difficulty levels in late adolescents. These earlier findings therefore raise the possibility that cognitive performance sensitivity to peer presence may increase as children get older, peak during adolescence, a period of life in which peer relationships take on a heightened importance compared to childhood [71,72,73], and then stabilize in adulthood. In this hypothesis, SFI developmental trajectory could thus follow the same inverted U-shaped developmental pattern as that observed for reward-related behaviors [67]. Here, children experienced social facilitation, with no evidence of a quantitatively greater sensitivity to peer presence than adults (no Age × Social condition interaction). Effect sizes were, however, larger in children than in adults. Cohen’s d_s_, for example, reached medium effect sizes of 0.61 and 0.51 in children (for numerosity and phonological comparisons, respectively), compared to small effect sizes of 0.38 and 0.21 in adults. Testing the present paradigm in a greater variety of school ages is therefore needed to determine whether the present difference in effect size represents the very early stage of a developmental difference peaking during adolescence. 

The past fifteen years or so saw a tremendous interest in using findings from neuroscience and cognitive psychology to inform the practice of education [74]. It was argued that, within certain limits, neuroscience can be used to guide cognitive psychology, which, in turn, can be used to guide education [75]. The present study is part of this ongoing effort. Its results are not sufficient to make any pedagogical recommendation, but they hold out hope that a comprehensive understanding of when peers’ mere presence benefits or, on the opposite, interferes with education could ultimately find real-life applications in the classroom. We show here that a familiar peer’s presence improves the way 4th-graders deal with mastered tasks. This finding provides some support to pedagogical methods dedicating collective, in-class time to the practice of already learned skills [76]. It also raises the possibility, to be tested in future studies, that the positive experience of outperforming peers in mastered tasks could have spill-over benefits on other tasks by enhancing pupils’ self-confidence [77]. Education, however, is mainly about developing new skills to master novel tasks. It is therefore highly vulnerable to the negative effects of peers’ presence. Such negative influence was already demonstrated for complex (relational) reasoning [13], and is likely to affect other complex cognitive skills. Fraction learning, for example, is a notoriously difficult mathematical skill to master [78]. To what extent does a peer’s mere presence make understanding fractions even more difficult? Peer presence often goes together with social comparison [79,80]. Comparing oneself to a slightly better coactor can improve cognitive performance when one feels close to the coactor and believes in one’s ability to achieve the task [81,82]. Could creating the conditions of a beneficial social comparison counteract social inhibition and ease fraction learning? These are but a few examples of the questions that can be addressed experimentally to gain knowledge potentially useful to educators. Meanwhile, though, the present behavioral study, based on a paradigm easily transferable to the scientific context of a MRI scanner, paves the way towards a neuroscience investigation of the mechanisms mediating peer presence effects in education and their evolution across development. 

## 5. Conclusions

We found that a schoolmate’s presence enabled children to perform mastered numerosity and phonological comparisons more like adults, with a better response strategy and faster and less variable response times than children tested alone. These findings confirm the influence of peer presence on skills relevant to education and stress the need for additional studies including older individuals and nonmastered tasks. They also lay the groundwork for exploring how the brain mechanisms mediating this fundamental social influence evolve during development.

## Figures and Tables

**Figure 1 biology-10-00902-f001:**
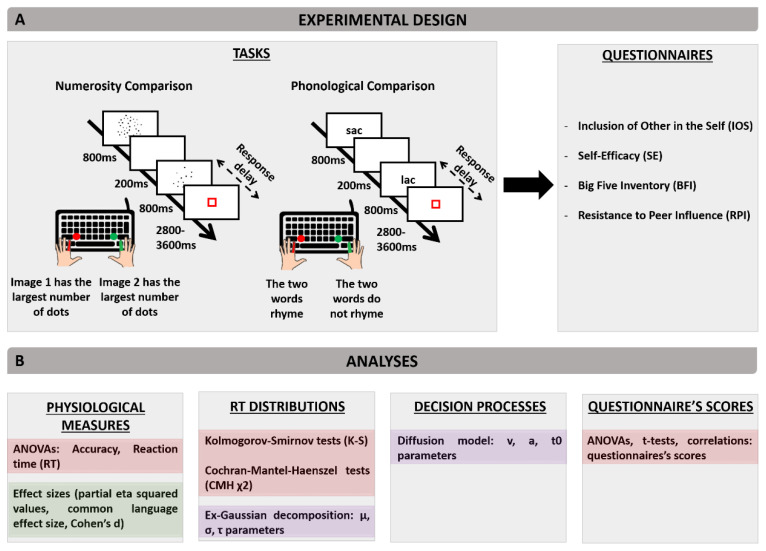
Schematic illustration of experimental design (**A**) and overview of conducted analyses (**B**). The task mixed two trial types (numerosity and phonological comparisons), and was followed by com-pletion of 4 questionnaires. Statistics are given in red boxes, effect sizes in green boxes, and computa-tional analyses in purple boxes.

**Figure 2 biology-10-00902-f002:**
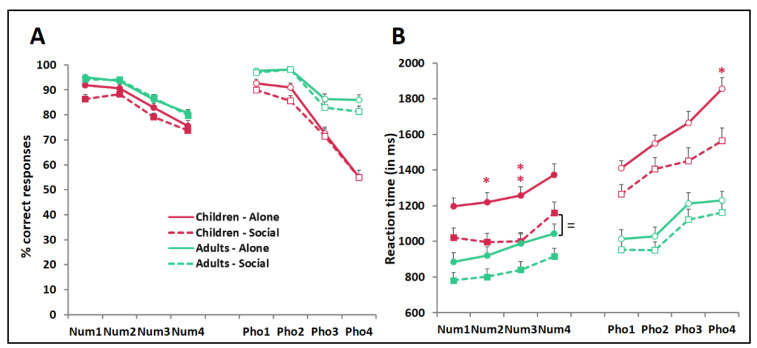
Accuracy (**A**) and reaction time (**B**) for numerosity (Num) and phonological (Pho) comparisons of increasing difficulty (Levels 1–4 for both Num and Pho) in children (pink) and adults (green) tested with (Social, dashed lines) *vs.* without (Alone, solid lines) a peer. Data points are means + SEM. Symbols denote results from Student’s *t*-tests with the Bonferroni adjustment for multiple comparisons the asterisks indicate differences between the Social and Alone conditions (* *p* < 0.05, ** *p* < 0.01), the = sign underscores the lack of difference between the reaction times of children tested with a peer and those of adults tested alone for all four Num difficulty levels.

**Figure 3 biology-10-00902-f003:**
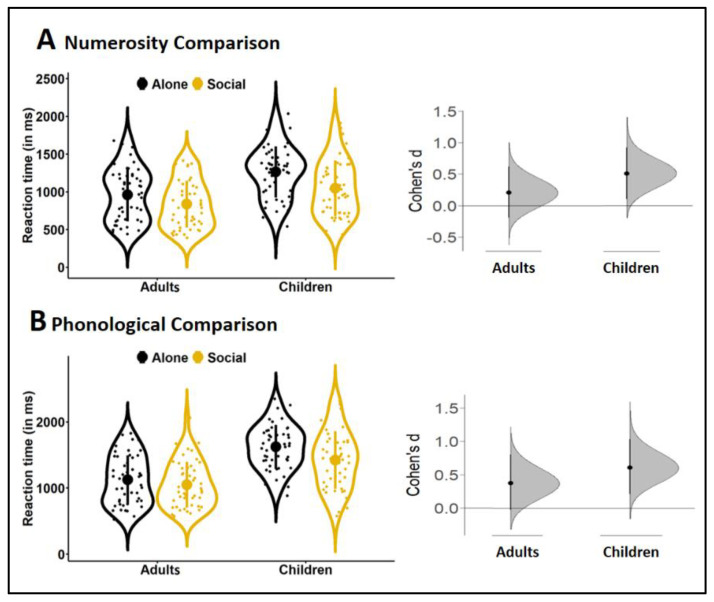
Average performance across difficulty levels for numerosity (**A**) and phonological (**B**) comparisons in adults and children tested without (Alone, black) *vs.* with (Social, yellow) a peer. For each trial type, reaction times are illustrated by the violin plots (left) with the large dot indicating the mean, the vertical line the standard deviation, the small dots the individual data, and the surrounding shape the density of the data. The corresponding effect sizes (Cohen’s d_s_) are presented as bootstrap 95% confidence intervals (right).

**Figure 4 biology-10-00902-f004:**
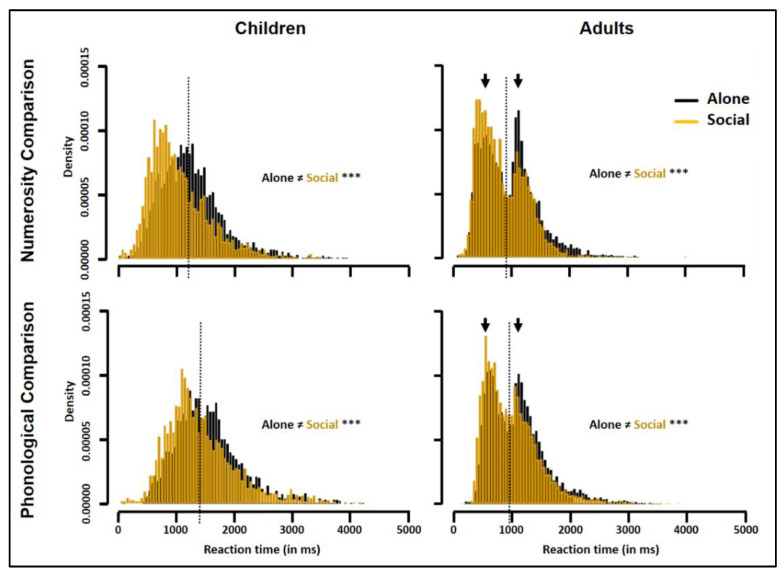
Group RT distribution in children (**left**) and adults (**right**) for numerosity (**top**) and phonological (**bottom**) comparisons in the Social (yellow) compared to the Alone (black) condition. *** denote differences between the Alone and Social conditions as revealed by K-S tests (*p* < 0.001). The arrows point to the 1st and 2nd peaks of the adults’ bimodal RT distribution. The dotted lines show the RT values used to separate fast from slow responders (see Figure 6).

**Figure 5 biology-10-00902-f005:**
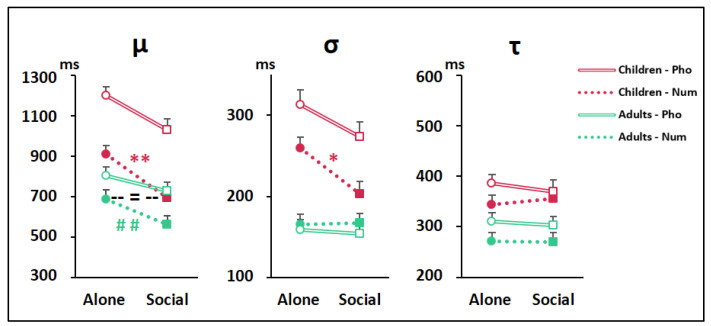
Ex-Gaussian decomposition of RTs into average performance (**µ**), variability in performance (**σ**), or extremes in performance (**τ**). Data points are means + SEM. Symbols denotes results from Student’s *t*-tests with the Bonferroni adjustment for multiple comparison: the asterisks indicate differences between the Social and Alone conditions (* *p* < 0.05, ** *p* < 0.01). The = sign underscores the lack of difference between children tested with a peer and adults tested alone for the µ parameter.

**Figure 6 biology-10-00902-f006:**
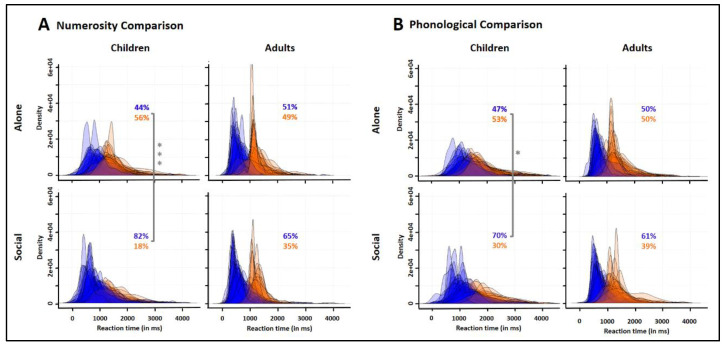
Superimposed individual RT distributions classifying children and adults into fast responders (with an early RT peak; blue) *vs.* slow responders (with a late RT peak; orange) for numerosity (**A**) and phonological (**B**) comparisons in the Social (bottom) compared to the Alone (top) condition. The proportion (in %) of both responder types are given on each graph. Peer presence increased the proportion of fast responders (blue percentages) in children more than it did in adults. Asterisks indicate differences between the Social and Alone conditions as revealed by CMHχ^2^ tests: *** χ^2^(1) = 13.81, *p* < 0.001, * χ^2^(1) = 4.52, *p* = 0.03.

**Figure 7 biology-10-00902-f007:**
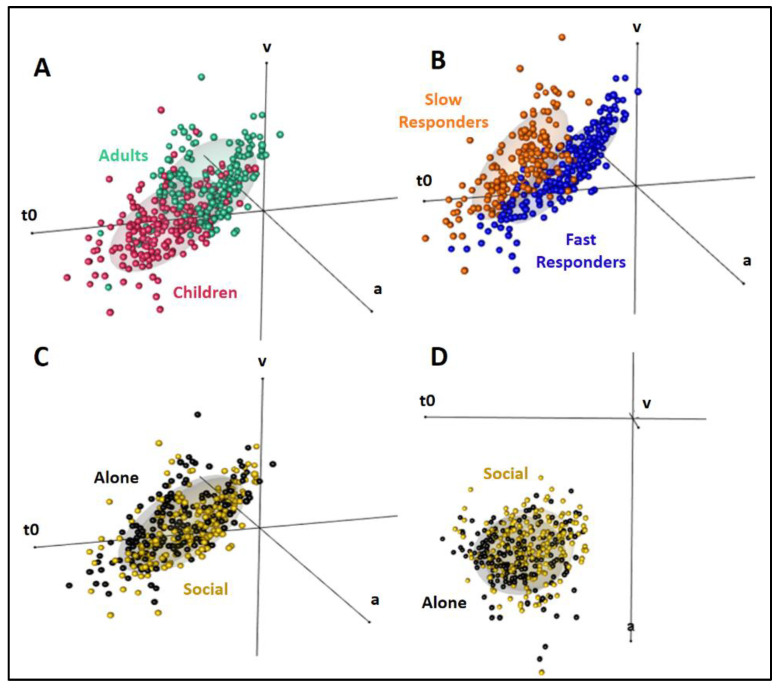
3-D representation of the diffusion model parameters v (drift rate), a (threshold), and t0 (non-decision time) Figure 3. axis representation. (**A**): Adults (green) *vs.* children (pink): v was higher, while a and t0 were lower in adults than children. (**B**): Fast responders (blue) *vs.* slow responders (orange): Only t0 changed, being lower and less variable in fast responders. (**C**,**D**): Alone (black) *vs.* Social (yellow) conditions illustrated from two different perspectives. Only t0 changes, being lower in the Social than in the Alone condition.

## Data Availability

Data supporting reported results can be found on the Open Science Framework website at https://osf.io/pj5r6/ (accessed on 30 July 2021).

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
