# Peer review of "Peer Presence Effect on Numerosity and Phonological Comparisons in 4th Graders: When Working with a SchoolMate Makes Children More Adult-like"

_biology, 2021, doi:10.3390/biology10090902_

Round 1

Reviewer 1 Report

The manuscript reports one single study (with a satisfactory statistical power) investigating the peer presence effects on fourth-graders in comparison to young adults. Due to the investigated population, the research is original and is of high-interest for the community of social psychologists. Hopefully, publishing this research in an outlet such as Biology could also attract the attention of a broader audience and could stimulate interdisciplinary approach of the peer presence phenomenon. The other major strength of the present research is the sophisticated and impressive data analyses that the authors led, as compared to what is usually done in the field of mere presence effects. I believe that the present manuscript could stand as an example of data analyzing strategy that could be more systematically conducted in other research investigating socio-contextual effects on cognitive performance.

That being said, I have one concern regarding the implication of the findings. The authors claim that the developmental exploration of peer presence effects on academic achievement has the potential for informing educators. In the long run, I also believe it might but for now, I would recommend the authors to moderate their rhetoric, for several reasons:

  • Results showed that peer presence improves the way 4th-graders deal with mastered tasks. But at the very least, there is little or no evidence that this would be true for skills that are not fully acquired yet. And there is ample evidence in the literature (as stated by the authors themselves) that it could even have opposite effects on non-mastered tasks. Yet this is what is expected from pupils at school: mastering new tasks by developing new skills. As a result, whether peer presence is beneficial in school settings remains unsure. It could well be that the experience of ease that pupils may have during peer presence performance on mastered tasks contributes to enhancing self-confidence, which in the end may indirectly be beneficial to the acquisition of new skills. But further research is needed to investigate the spillover effects of the positive experience they may gain from outperforming mastered tasks in peer presence.
  • Previous research on SFI effects and related effects has identified many moderators of peer presence effects. It is well-accepted that peer presence often goes together with social comparison. Individuals tend to engage in a comparison with the present peer (see for example Huguet, Galvaing, Monteil & Dumas, 1999). Whether such comparison will result in improved or deteriorated cognitive performance also depends on other factors, such as the nature of the task and the beliefs individuals hold about the incremential or fix nature of the subtended skills (see for example, Normand & Croizet, 2013) as well as the psychological closeness they feel with the target comparison (see Tesser’s self-evaluation maintenance model, 1988). As far as pupils are concerned, what may also come into play is their individual academic history of success vs. failure. All in all, in the present reported study, participants are 4th graders whose academic history is rather short and therefore probably only has a negligeable moderating role in the observed outcomes. Moreover, the authors chose to focus on dyads with familiar peers. As a consequence, we simply do not know what the results would be with slightly older pupils who have longer academic history and neither we know what would happen with unfamiliar peer.
  • The present study is well-designed and I found quite impressive all the analyses conducted. However, this is a single study that would need to be, at the very least, replicated before even thinking of making practical and pedagological recommendations.

I recommend the authors to temper their perspectives (especially in the last paragraph of the discussion) by acknowledging that the present findings do not inform us on peer presence outcomes with older pupils (i.e., after a couple of years of positive or negative academic experience), on the potential social comparison effects that can arise from peer presence and on peer presence effects on completely new tasks.

Author Response

Reviewer 1

We appreciate the positive feedback of the Reviewer on our analyses and wish to thank her/him for that and for giving us the opportunity to share our results with the broad audience of Biology. We have tempered our inferences about education throughout the manuscript.

The Simple summary now reads:

Future studies pursuing this hitherto neglected developmental exploration of peer presence effects on academic achievements might have the potential to help educators tailor their pedagogical choices to maximize peer presence when beneficial and minimize it when harmful.

The Introduction now reads:

Applying this approach to children could unveil both the positive and negative influences of peers' mere presence on education. This knowledge might ultimately provide useful insights to educators about when to minimize, or on the contrary, maximize peers presence.

The last paragraph was entirely re-written. It now reads:

The past fifteen years or so have seen a tremendous interest in using findings from neuroscience and cognitive psychology to inform the practice of education [76]. It has been argued that, within certain limits, neuroscience can be used to guide cognitive psychology, which, in turn, can be used to guide education [77]. The present study is part of this ongoing effort. Its results are, of course, not sufficient to make any pedagogical recommendation, but they hold out hope that a comprehensive understanding of when peers' mere presence benefits or, on the opposite, interferes with education could, ultimately, find real-life applications in the classroom. We show here that a familiar peer's presence improves the way 4th-graders deal with mastered tasks. This finding provides some support to pedagogical methods dedicating collective, in-class time to the practice of already learned skills [78]. It also raises the possibility, to be tested in future studies, that the positive experience of outperforming peers in mastered tasks could have spill-over benefits on other tasks by enhancing pupils' self-confidence [57]. Education, however, is mainly about developing new skills in order to master novel tasks. It is therefore highly vulnerable to the negative effects of peers' presence. Such negative influence has already been demonstrated for complex (relational) reasoning [13], and is likely to affect other complex cognitive skills. Fraction learning, for example, is a notoriously difficult mathematical skill to master [79]. To which extent does a peer's mere presence make understanding fractions even more difficult? Peer presence often goes together with social comparison [80,81]. Comparing oneself to a slightly better co-actor can improve cognitive performance when one feels close to the co-actor and believes in one's ability to achieve the task [82,83]. Could creating the conditions of a beneficial social comparison counteract social inhibition and ease fraction learning? These are but a few examples of the questions that can be addressed experimentally in order to gain knowledge potentially useful to educators. Meanwhile, though, the present behavioral study, based on a paradigm easily transferable to the scientific context of a MRI scanner, paves the way towards a neuroscience investigation of the mechanisms mediating peer presence effects in education and their evolution across development.

Reviewer 2 Report

This is an extremely valuable study investigating (1) the presence of familiar peers on performance of 4th-grader children in tasks measuring literacy and numeracy, two tasks relevant to education in elementary school and (2) comparing the effects of peer presence on performance of children to that of college-aged young adults.  What makes this study noteworthy and original is th use of a large sample size (111children and 100 adults) and the examination for the first time of whether the effects of peer presence on cognitive performance largely described in adults can also occur in children. A second important aspect of this study was the selection of the cognitive tasks for the children not only because they were similar to those given to the adults, making comparisons across ages very conclusive, but also because the simplification of the tasks for the children made them of equivalent difficulty across both ages.  The data provide evidence that children performed faster in the social situation than alone in both tasks and this positive effect was as important in children than that seen in colle-aged adults.

The text is clearly written and the methods provide enough details to allow replication.  The results have been carefully and systematically analyzed and the figures and graphs illustrate with clarity the main findings.  The expertise of the authors and volume of work invested in the data analyses makes the conclusions truly compelling.  The discussion offers important interpretations of the implication of the results in relation to data in the literature as well as for the potential of the results to inform educators in the selection of their pedagogic methods for best academic achievements.  Overall, this manuscript is at the utmost importance not only for the novel basic knowledge it provides in the field, but because it also paves the way to future studies looking at the developmental trajectories of the effects of social presence in cognitive performance from childhood to adolescence using the same cognitive tasks as well as allowing to use neuroimaging tools to better understand the neural mechanisms governing this social effect.  Thus, the data have a unique value to educators, psychologists as well as neuroscientists.

I have only one comment to the authors: In the Method section on page 4 lines 148-151. the authors note the use of 288 trials for the adults but only 144 trials for the children. What was the reason for this difference? Could it be possible that the smaller number of trials for the children may have reduced their performance on the phonological task as compared to adults, expecially at the most difficut part of this task (see Figure 2A, left graph)?

Author Response

We would like to thank Reviewer 2 for taking the time to evaluate our study and for his/her very positive feedback that we appreciate.

It is indeed possible that children could have achieved better performance had they been tested with as many trials as the adults. But, as teachers kindly adapted their day schedule to allow all their pupils to each, in turn, participate in the experiment, we tried to keep the time spend by each child outside the classroom to the minimum possible.

The Methods section now states:

The task was sized down to 144 trials for children in order to minimize the duration of the interruption of their school day.

Reviewer 3 Report

The manuscript is well written and documented. The title summarizes in an attractive way the study presented.

The arguments for conducting such a research are well presented in the introduction, but it would be nice to include some more references to similar studies (with similar methods used) in the introductory part.

Chapter 2 is quite hard to be read....I would suggest to present the ideas in a graphical way: to include a short model/plan/figure/etc of the research methodology, with the methods & tools used in each step of the research.

Also, I would suggest to include in the last chapter some more practical recommendations regarding the utility of the study's findings.

Otherwise, I appreciate your paper and the work performed and I recommend the publication of the manuscript! 

Author Response

We would like to thank Reviewer 3 for reviewing our manuscript and for giving us very constructive and positive comments.

Similar studies are cited in the Introduction:

Studies notably aimed to understand the negative influence of peers on adolescents’ reasoning [13,20] and decision-making [21–24].

We have modified Figure 1 which now provides an overview of all measures and analyses.

As evoked above in response to Reviewer 1, the last paragraph of the Discussion on the relevance of the study to education has been entirely re-written.

Reviewer 4 Report

Although the article is scientifically well organized and an extensive data analysis has been done, my recommendation for Reject is based on the following aspects that cannot be corrected:

  • The article does not fit the strict scope of the journal (even if Biology does naturally encompass all the processes - including the cognitive and social ones - of living beings). There are other journals (namely on the Psychology domain) where this article would fit much better so the recommendation is to transfer the article to one of those journals.
  • Starting with the title, the ambition of the article is not supported by the study. The fact that the study was casuistic (one time experiment) does not allow to establish long term inferences  (as claimed, for instance, in the first sentence of the abstract: "The present study explores the potential impact of peers' omnipresence at school on children's academic performance"). All it is possible to assess from the study is the difference to the individual reaction to a test or challenge according to the presence or not of a peer. It is not possible to extract any conclusions related to cognitive development, working abilities, development of skills, and so on as claimed by the authors.

Author Response

We thank the Reviewer for his/her time and helpful comments. At Reviewer 1's request, we have adjusted the wording of our inferences about education throughout the manuscript. In answer to Reviewer 4's concerns, we have also corrected the title and abstract.

The title now refers to the tasks rather than to numeracy and literacy in general: "Peer presence effect on numerosity and phonological comparisons in 4th graders: when working with a schoolmate makes children more adult-like."

The first sentence of the abstract is now more specific, and reads: "Little is known about how peers' mere presence may, in itself, affect academic learning and achievement. The present study addresses this issue by exploring whether and how the presence of a familiar peer affects performance in a task assessing basic numeracy and literacy skills: numerosity comparison and phonological comparison."